



# A D-vine copula-based quantile regression towards merging satellite precipitation products over a rugged topography at the upper Tekeze Atbara Basin of the Nile Basin

Mohammed Abdallah[1,2,7], Ke Zhang[1,2,3,4,5,6], Lijun Chao[1,2,4], Abubaker Omer[8], Khalid Hassaballah[9], Kidane Welde Reda[10,11], Linxin Liu[1,2], Tolossa Lemma Tola[1,2], Omar M. Nour[1,2,7]

[1]Key Laboratory of Hydrologic-Cycle and Hydrodynamic-System of Ministry of Water Resources, Hohai University, Nanjing, Jiangsu, 210024, China
[2]College of Hydrology and Water Resources, Hohai University, Nanjing, Jiangsu, 210024, China
[3]Yangtze Institute for Conservation and Development, Hohai University, Nanjing, Jiangsu, 210024, China
[4]China Meteorological Administration Hydro-Meteorology Key Laboratory, Hohai University, Nanjing, Jiangsu, 210024, China
[5]The National Key Laboratory of Water Disaster Prevention, Hohai University, Nanjing, Jiangsu, 210024, China
[6]Key Laboratory of Water Big Data Technology of Ministry of Water Resources, Hohai University, Nanjing, Jiangsu, 210024, China
[7]The Hydraulics Research Station, PO Box 318, Wad Medani, Sudan.
[8]Korea Advanced Institute of Science & Technology (KAIST), Daejeon, South Korea.
[9]IGAD Climate Prediction and Applications Center (ICPAC), Nairobi, Kenya.
[10]Key Laboratory of Water Cycle and Related Land Surface Processes, Institute of Geographic Sciences and Natural Resources Research, Chinese Academy of Sciences, Beijing, China.
[11]Tigray Agricultural Research Institute, Mekele, Ethiopia.

*Correspondence to*: Ke Zhang (kzhang@hhu.edu.cn)

**Abstract.** Precipitation is a vital key element in various studies of hydrology, flood prediction, drought monitoring, and water resources management. The main challenge in conducting studies over remote regions with rugged topography is that weather stations are usually scarce and unevenly distributed. However, open-sourced satellite-based precipitation products (SPPs) with the suitable resolution provide alternative options in these data-scarce regions, typically associated with high uncertainty. To reduce the uncertainty of individual satellite products, we have proposed a D-vine Copula-based Quantile Regression (DVQR) model to merge multiple SPPs with rain gauges (RGs). DVQR model was employed during the 2001-2017 summer monsoon seasons and compared with two other quantile regression methods based on the Multivariate Linear (MLQR) and the Bayesian Model Averaging (BMAQ), respectively, and two traditional merging methods: the simple modeling average (SMA) and the one-outlier-removed average (OORA) using the descriptive and categorical statistics. The rugged topography region of the Upper Tekeze-Atbara Basin in Ethiopia was selected as the study region. The Results indicated that the precipitation data estimates with DVQR, MLQR, and BMAQ models and traditional merging methods outperformed the downscaled SPPs. Monthly evaluations reveal that all products perform better in July and September than in June and August due to precipitation variability. DVQR, MLQR, and BMAQ models exhibit higher accuracy than the traditional merging methods over UTAB. The DVQR model substantially improved all the statistical metrics considered over





BMAQ and MLQR models. However, DVQR model does not outperform BMAQ and MLQR models in the probability of detection (POD) and false alarm ratio (FAR), although it has the best frequency bias index (FBI) and critical success index (CSI) among all the employed models. Overall, the newly proposed merging approach improves the quality of SPPs and

demonstrates the value of the proposed DVQR model in merging multiple SPPs over rugged topography regions such as UTAB.

## 1 Introduction

Optimizing water resource management requires accurate and reliable meteorological information at fine spatial and temporal resolution. Precipitation is vital in various studies, such as weather forecasts, hydrology, agricultural practices,

flood prediction, drought monitoring, and water resources management (Kimani et al., 2017; Sun et al., 2018; Amjad et al., 2020; Zhang et al., 2016). However, due to the lack of rain gauges, the rugged topography, and the significant spatial variability in precipitation, accurate precipitation estimation in remote areas such as the Nile River Basin is extremely difficult (Kimani et al., 2017). Accurate high spatiotemporal resolution precipitation data in regions with rugged topography are precious for hydrological simulation and extreme event investigations. However, precipitation data contain significant

uncertainty due to the limitation in precipitation recording and estimation methods (Alfieri et al., 2014; Qi et al., 2019).
The conventional precipitation data from the rain gauges lack sufficient spatial and temporal resolution, particularly in the rugged topography (Yong et al., 2010; Ma et al., 2015; Kidd et al., 2017). In tropical climates, at least one gauge in 600-900 $km^2$ of flat areas and 100-250 $km^2$ of mountain regions is recommended for ground precipitation measuring networks (Wmo, 1994), but this criterion was usually not met in practice (Worqlul et al., 2017). The Upper Tekeze basin (UTAB), located in a

tropical region, is one of the major tributaries of the transboundary Nile River (See Fig. 1). It has inadequacy in the rain gauge network, with only one gauge per 1400 $km^2$ (Gebremicael et al., 2019). The main difficulty in capturing the spatial precipitation variability in this basin lies in the uneven distribution of meteorological stations (Belete et al., 2020). Alternate precipitation data from satellite precipitation products (SPPs) are highly desirable for extreme precipitation estimates (Barrett and Martin, 1981). Although the use of SPPs at regional and global scales has increased substantially over recent years

(Belete et al., 2020; Reda et al., 2021), the quality of SPPs over a complex topography is still problematic since these SPP products are significantly influenced by climate conditions, seasonal variability, precipitation type, and complex topography (Kidd and Huffman, 2011; Hou et al., 2014). SPPs data were used as input to the hydrologic modeling for simulations of extreme flood events (Li et al., 2015; Fenta et al., 2018; Muhammad et al., 2018b). While these studies highlighted the capability of SPPs in flood modeling, they also reported inherent uncertainties in SPPs (Zubieta et al., 2017), such as the

under-/over-estimation of SPPs, which may lead to high uncertainties in streamflow simulation and drought monitoring (Reda et al., 2022; Gebremicael et al., 2022; Reda et al., 2021). The quality of individual SPPs is improved at longer time\scales (monthly to daily); they still encounter several inevitable errors such as over/underestimations, indicating that there is still potential for further enhancement of their performance.





Several efforts have been made to increase the accuracy of precipitation estimates with SPPs, including improvements in
calibration methods, bias correction, and merging multiple SPPs (Chao et al., 2018; Rahman et al., 2018a; Kumar et al.,
2019; Muhammad et al., 2018a; Sun et al., 2016). The merging approach is a concept of blending multiple products into a
single new product (Rahman et al., 2018b). The merging techniques were employed recently throughout many statistic
approaches, substantially increasing the accuracy of hydrological models (Raftery et al., 2005b). The merging procedures of
multiple SPPs produce a single source of precipitation data, characterized by the highest performance than all or a majority
of individual SPPs and demonstrated their abilities in hydrological applications and extreme events (Rahman et al., 2021;
Rahman et al., 2020b). The most commonly traditional merging methods are simple model averaging (SMA), one-outlier-
removed average (OORA), inverse error variance weighting (IEVW), and the optimized weight average (OWA). These
methods indicate light improvement in accuracy compared to original SPPs (Shen et al., 2014; Yumnam et al., 2022).
Therefore, a bias-correction of errors is needed to improve the quality and spatial distribution of precipitation data.

Recently, merging multiple satellites, reanalysis, explanatory variables, and ground products has opened up new possibilities
to improve the estimation of precipitation data across scarce regions at all temporal scales (Mastrantonas et al., 2019).
However, the merging approaches to create a new product are still in their early stages. A few research explored different
techniques for merging multiple SPPs from various sources, such as Geographically Weighted Regression (GWR) (Chao et
al., 2018), Stepwise Regression (STER) (Xiao et al., 2020), Bayesian model averaging (BMA) (Ma et al., 2018), Wavelet
Transform Analysis (Pradhan et al., 2015), and Kriging-based algorithms (Manz et al., 2016). The efficacy of these merging
techniques in raising precipitation estimate quality has been demonstrated. However, most of them are based on strong (ad
hoc) hypotheses that might not be accurate in practice (Wu et al., 2020). The dynamic and cluster BMA reflect the highest
potential and capabilities to produce high-quality merged precipitation data and performed better than traditional merging
methods (e.g., IEVW, OWA, and OORA) and row satellite data (e.g., Multi-Source Weighted-Ensemble Precipitation-
MSWEP) in the Tibetan Plateau-China (Ma et al., 2018), Pakistan (Rahman et al., 2020c; Rahman et al., 2020a), and
Vamsadhara River basin-India (Yumnam et al., 2022). But the model produces a combined single-value prediction by
averaging the deterministic model outputs linearly, which does not accurately reflect the contributions of each input variable
(Jennifer et al., 1999). Several Machine Learning (ML) developed to merge multiple satellite products, such as Random
Forest (Nguyen et al., 2021a), Multilayer Perceptron Neural Network (Kolluru et al., 2020), Support Vector Machine
(Kumar et al., 2019), and Quantile Regression Forests (Bhuiyan et al., 2018; Bhuiyan et al., 2019); which reflect their ability
to capture the nonlinear relationship between the variables. Therefore, whether these approaches affect the spatiotemporal
scales of pattern distribution of precipitation data is unclear.

The copula approach has proven successful in hydrometeorological applications for modeling the multivariate nonlinear
interdependence of input data using the joint, marginal distributions. Successful applications of the bivariate copulas in
correcting the error of satellite precipitation products have been reported. For instance, (Sharifi et al., 2019) employed a t-
copula approach to adjust the additive errors to improve SPP quality. The multivariate Gaussian copula approach was
utilized to reduce the uncertainty of precipitation data for the bias correction of two SPPs (Moazami et al., 2014). The D-vine





Copula-based Quantile Regression (DVQR) algorithm was introduced by Kraus and Czado (2017) to predict quantile conditional with the highest flexibility. The DVQR algorithm demonstrated a high ability and potential to capture the nonlinear relationships between the variables in different hydrology applications such as reference evapotranspiration estimation (Abdallah et al., 2022), soil moisture modeling (Nguyen et al., 2021b), and drought prediction (Wu et al., 2022). To merge multiple individual SPPs with ground stations and consider the geographical and meteorological factors, the DVQR algorithm may be suitable to model the dependence structure of the variables, which can provide a robustness model. Here in this present study, we aim to merge daily precipitation data from multiple individual satellite-based precipitations (SPPs) with rain gauges (RGs) and meteorological and topographical variables over the upper Tekeze-Atbara basin (UTAB). Multiple SPPs have been employed in this study, such as Tropical Applications of Meteorology using SATellite (TAMSAT v3.1), the Climate Prediction Center MORPHing Product Climate Data Record (CMORPH-CDR), Global Precipitation Measurement (GPM) Integrated Multi-satellite Retrievals for GPM (IMERG v06) and Precipitation Estimation from Remotely Sensed Information using Artificial Neural Network (PERSIANN-CDR). The explanatory variables of meteorological and topographical are wind speed (WS), elevation (DEM), slope (SLP), aspect (ASP), hill-shade (HSHD), and surface soil moisture (SSM). We suggest using the DVQR model for the first time to merge daily precipitation data during the summer monsoon (June, July, August, and September) from 2001 to 2017. We further compared the performance of the DVQR model with MLQR and BMAQ models and two traditional merging methods (SMA and OORA) using descriptive and categorical statistics.

## 2 Material and methods

### 2.1 Study area

The Tekeze Atbara Basin (TAB) is one of the main tributaries of the Nile River as shown in Fig1. The current study was conducted at the Upper Tekeze Atbara Basin (UTAB), which is located in the northwestern part of Ethiopia, between longitudes 37° 30′ 0" – 39° 48′ 0" E and latitudes 11° 30′0" – 14° 18′ 0" N, with a contributing area of approximately 45,694 km$^2$, with more than 50% of the total area located at an elevation from 2000 to above 3000 m.a.s.l. The TAB contributes 13% of the entire Nile Basin area and 14% of the annual flow at the High Aswan Dam in Egypt (Gebremicael et al., 2019). A complex topography of the basin characterizes by significant variation of elevation from 833 to 4530 m.a.s.l, based on topographic information of the Digital Elevation Model (DEM).

The basin is characterized by a semi-arid climate in its northern and eastern parts and a semi-humid characteristic climate in the southern region. Precipitation over the basin occurs from June to September, accounting for more than 70% of annual precipitation, ranging from 400 mm/yr in the eastern part to 1200 mm/yr in the southwestern parts of the basin (Gebremicael et al., 2019). The mean annual temperature over the basin ranges from 11ºC to 31ºC in the eastern mountain region to the western lowlands, and the highest mean monthly temperature accurses in May, and the lowest is in December.

<Figure. 1, inserted here, please>





## 2.2 Data collection and processing


Rain gauges (RGs) and satellite-based precipitation data from CMORPH-CRD, IMERG v06, TAMSAT v3.1, and PERSIANN-CDR were gathered over 17 years, during summer monsoon (June, July, August, and September) from January 2001 to December 2017.

### 2.2.1 Rain gauge data

The daily precipitation data from ten rain gauging stations from January 2001 to December 2019 was provided by the Ethiopian National Meteorological Agency (NMA). Most of the RGs are localized in complex topography in the northeastern part of TAB; however, they are very sparse in the western, as present in Fig.1. Moreover, Table 1 summarizes the geographical locations of rain gauge, elevation, and statistical data characteristics. There are more than 75 rain gauge observations across the upper TAB. However, most of these gauges have more than 50% missing records, and some of the

gauging stations went out of service (Gebremicael et al., 2019).

<Table 1, inserted here, please>

### 2.2.2 Remote sensing data

CMORPH is another NOAA-CPC product. In contrast to other items, the CMORPH product does not utilize an algorithm to combine passive microwave (PMW) and infrared (IR) estimates but instead utilizes IR information to predict the temporally

and spatial evolution of clouds rather than rainfall estimations (Joyce et al., 2004). It employs precipitation developing a sense from low orbit PM measurements and propagates these characteristics using both temporal and spatial pixel density IR data.   This method is highly adaptable because it allows adding any precipitation estimate using PM satellites. The CMORPH-CRD was used in this study.

The GPM IMERG V06 produces precipitation with a fine spatial resolution (0.1º) and half-hourly temporal resolution

between 60º S and 60º N. The Dual-frequency phased array Precipitation Radar (DPR). A multi-channel GPM Microwave Imager (GMI) data are used to validate and integrate precipitation estimates from different PMW satellites to create precipitation using the IMERG algorithms. Furthermore, the PERSIANN-CCS algorithm and morphing technique were employed to compute the precipitation rate from microwave-calibrated infrared IR and global gridded precipitation (Tan et al., 2019; Huffman et al., 2015). In this study, the GPM IMERG Final Run V06 was used.

The PERSIANN-CDR product estimates the precipitation by utilizing the Infrared Brightness temperature archive from GridSat-B1 (Hsu et al., 1997). The Global Precipitation Climatology Project (GPCP) version 2.2 product was integrated with estimated precipitation from the PERSIANN algorithm for bias correction at the spatial resolution of (2.5° × 2.5°) and spatial covers 60N – 60S from 1983 to the present for daily, monthly, and yearly. The present study utilized the PERSIANN-CDR product, which was downloaded free of charge from the Center for Hydrometeorology and Remote Sensing (CHRS).



The University of Reading for Africa established the TAMSAT, based on Thermal Infrared Imagery from Meteosat satellite and observation gauges with fine spatial resolution 0.0375 approximately (4km) from 1983 to the present for daily, pentadal, decadal, and monthly (Maidment et al., 2014; Maidment et al., 2017). The TAMSAT product version 3.1 was utilized in the present study.

### 2.2.3 Explanatory variables

The Digital Elevation Model (DEM) data employed in this study was obtained from Shuttle Radar Topography Mission (SRTM) with a spatial resolution (90m) and rescaled to 1 km using Bilinear Interpolation (BIL) Techniques. In contrast, the topographic variables were derived from DEM, including slope (SLP), aspect (ASP), and hill-shade (HSHD), as shown in Fig.2. The ASP characterized across the basin from -1º to 358.3º, whereas the SLP range from 0.0º to 45.7º, and HSHD from 8 to 254.

Daily 10-meter wind speed (WS) at 0.25º spatial resolution was obtained from the ERA5, the fifth generation of the European Center for Medium-Range Weather Forecasts (ECMWF). In contrast, the daily surface soil moisture (SSM) was obtained from Global Land Evaporation Amsterdam Model (GLEAM 3.6a) with a spatial resolution of 0.25º during the summer monsoon (JJAS). Recently, some investigations indicated that using WS and SSM can improve the estimation of SPPs in various regions (Chao et al., 2018; Kumar et al., 2019). Further, the BIL techniques were applied to downscale the

WS and SSM from coarse resolution (0.25º) to the fine resolution of 0.01º. The average WS across the basin ranges from 6.4 m/s to 9.4 m/s, while the SSM ranges from 0.24 $m^3/m^3$ to 0.39 $m^3/m^3$, as presented in Fig.2e and f.

<Figure. 2, inserted here, please>

### 2.3 Methods

Fig.3 presents the workflow of the merged SPPs approach developed in this study. First, the BIL technique is applied to

downscale original SPPs from coarse to fine spatial resolution (0.01º) during the summer monsoon from 2001 to 2017. Second, three DVQR, MLQR, and QBMA models were employed to merge downscaled SPPs with RGs and coupled with explanatory variables over the UTAB. The following is a more detailed description.

<Figure. 3, inserted here, please>

### 2.3.1 D-vine copula-based quantile regression (DVQR) model

Copulas are functions that integrate several univariate marginal distributions into single multivariate distribution, with all marginal distributions having the same uniform distribution on the [0,1] (Sklar, 1959; Nelsen, 2005). Copulas have previously been used to tackle complex issues in a variety of fields, including hydrology (Pham et al., 2016), engineering (Niemierko et al., 2019), and finance (Bouyé and Salmon, 2009). Most previous studies applied the copula approach in hydrology to model the dependence among two variables; however, only a few research studies have attempted to address

the issues associated with high dimensions. According to Aas et al. (2009), the vine copula, also known as Pair copula





construction considered a versatile technique for constructing a higher dimensional. Regular vine copulas are divided into two types: canonical (C-vine) and drawable vine (D-vine) copulas (Kurowicka et al., 2005); thus, each model explains how to decompose the density uniquely. The hierarchy of a D-vine copula is made up of nesting trees. Fig.4 represents a hierarchical tree D-vine structure with five variables.

200                                        <Figure. 4, inserted here, please>

To our knowledge, this study is the first to use D-vine copula-based quantile regression (DVQR) to merge multiple SPPs with RGs and coupled with explanatory variables across rugged topography like the UTAB. The DVQR model was first modeled using historical data and integrating numerous variables selected to estimate the conditional quantile. Compared to popular copula approaches such as elliptical copulas, parametric and Archimedean copulas, it offers dependency modeling of

high-dimensional and may represent nonlinear interactions among variables (Niemierko et al., 2019).

The ultimate focus of the DVQR model is to estimate the quantile of a randomly variable $Y \sim F_Y$ Given the output $x_1, \dots x_n$, $n$ of specific predictors $X_1, \dots X_n$ , $n > 1$ (Kraus and Czado, 2017). As follows, D-vines are utilized to simulate the joint distribution of $Y, X_1, \dots, X_n$ and compute the conditional quantile function of $Y$ given $X_1, \dots, X_n$ for $\alpha \in (0,1)$ as the inverted of the conditional distribution function:

$$q_\alpha(x_1, \dots, x_n) = F^{-1}_{Y|X_1,\dots,X_n}(\alpha|x_1, \dots, x_n),$$    (1)

where $V = F_Y(Y)$ and $U_j = F_j(X_j)$ are defined as independent variables with implementations $u_j = F_j(x_j)$. The right-hand side of Eq. (1) can be stated as follows using Sklar's theorem (Sklar, 1959):

$$F^{-1}_{Y|X_1,\dots,X_n}(\alpha|x_1, \dots, x_1) = F_Y^{-1}(C^{-1}_{V|U_1,\dots,U_n}(\alpha|u_1, \dots, u_n)),$$    (2)

Lastly, Eq. (1) can be modified to include the calculated marginals $\hat{F}_Y, \hat{F}_1, \dots \hat{F}_n$ and vine copula $\hat{C}_{V|U_1,\dots,U_n}$, as follows:

$$\hat{q}_\alpha(x_1, \dots, x_n) = \hat{F}_Y^{-1}(\hat{C}^{-1}_{V|U_1,\dots,U_n}(\alpha|\hat{u}_1, \dots, \hat{u}_n)),$$    (3)

where $\hat{u}_j = \hat{F}_j(x_j)$ represents the integral probability transformation computed using the continuous kernel smoother estimator (Parzen, 1962). The Onepar copula (ONC) family is selected for fitting copula selection because it is simple and flexible in terms of catching natural dependencies between hydrologic elements (Chen and Guo, 2019), which minimizes the computation costs when merging SPPs with fine spatial resolution. In the present study, simply five common ONCs are

used: the Gaussian (GA), Clayton (C), Frank (F), Gumbel (GU), and Joe (J) copulas.

To predict daily precipitation data, the DVQR model merges multiple SPPs using all data of the interdependence relationships among components. To reduce the high computational cost, we limited our testing to modeling at only five quantile levels (5th, 25th, 50th, 75th, and 90th). Using Eq. (3) and the parameters $\alpha = [0.05, 0.25, 0.50, 0.75, 0.90]$, conditional quantiles of the merged precipitation data were produced. Descriptive statistics, including CC, NSE, MAE, and

RMSE, were selected as key criteria to measure the reliability and sensitivity of merged precipitation data at various quantile levels.




### 2.3.2 Multivariate Linear-based Quantile Regression (MLQR) model

The MLQR model was proposed by Koenker and Bassett (1978), detailed description of the concept can be found in Koenker and Ng (2005). The method employs procedures equivalent to linear regression to compute the quantile levels of a
dependent variable based on predictor factors. The critical distinction between the MLQR model and linear regression is that minimizing is done based on conditional predicted quantile levels, while linear regression takes the conditional mean into consideration of the dependent variables. MLQR model is explained intuitively as fitting a linear model and bisecting the input so that 100 q% of the outputs are below the prediction values of the trained model. In practice, this is accomplished by training a linear model to the information and reducing the average quantile score.

### 2.3.3 Quantile Bayesian Model Averaging (QBMA) model

BMA is a technique that combines estimated forecast density from various models to generate a new prediction Probability Density Function (PDF). The predicted distribution of a merged precipitation data $x$, provided the observed rain gauges $X$ during the training phase and the independent estimates of $k$ models, can be stated using the theory of total probability as follows:

$$p(x|M_1, M_2, \ldots, M_{K=k}, X) = \sum_{i=1}^{k} p(M_i|X)p(x|M_i, X), \tag{4}$$

where $p(x|M_i, X)$ represents the posterior distribution of $x$ provided the predicted values $M_i$ and training dataset $X$. Moreover, $p(M_i|X)$ represents the likelihood of predicted data offered to the observed data $X$ during the training phase, which further indicates the weight of every model $M_i$. Hence the output of the BMA model is the mean weight of the predicted PDF produced from each model. Because the model predictions vary over time, Eq (4) can be phrased as:

$$p(x^t|M_1^t, M_2^t, \ldots, M_k^t, X) = \sum_{i=1}^{k} w_i p(x^t|M_i^t, X), \tag{5}$$

Note that w denotes the quality of the model throughout the training phase. To address this equation, it is generally acknowledged that such posterior distribution follows the Gaussian distribution with the average of observed data $f_i^t$ while the variance $\sigma_i^2$, related to SPPs, like that $p(x^t|f_i^t, X) \sim g(x^t|f_i^t, \sigma_i^2)$. BMA probabilistic modeling improves reliability by including weights from higher efficient precipitation products. BMA probabilistic modeling improves reliability by having
consequences from higher efficient precipitation products. It is essential to emphasize that for non-Gaussian forecasting variables (SPPs, RGs, and explanatory variables), a powerful transformation (e.g., Box-Cox) is employed to translate them both from their natural space toward a Gaussian space. The variability and weight of each prediction model can be determined using the log-likelihood formula. The Expectation-Maximization (EM) model was proposed by Raftery et al. (2005a) to optimize Eq. (6), which cannot be calculated analytically.

$$l(w_1, w_2, \ldots, w_k, \sigma^2) = log\left(\sum_{i=1}^{k} w_i . p(x|f_i, X)\right), \tag{6}$$





The EM algorithm execution instructions are laid out (Duan et al., 2007). The EM algorithm was employed to calculate unique weights ($w_k$). With an accurate assessment of weights for each precipitation product, it is simple to construct merged precipitation data using Eq. (4). The quantile-based BMA (BMAQ) technique transfers data from predictors to estimate the target at various quantile levels. In the present study, we suggest employing the BMAQ model to produce merged

precipitation data-based quantile levels and consider this for comparison with other models.

### 2.3.4 Traditional merging methods

This study adopted two commonly traditional merging methods of multiple SPPs, Simple Model Averaging (SMA) and one-outlier-removed average (OORA), as given in Eqs. (7) and (8), respectively.

$$R_{merg} = \frac{1}{n}\sum_{i=1}^{n} Sat_i ,$$
(7)

$$R_{merg} = \frac{1}{N-1}\sum_{i=1}^{n-1} Sat_i ,$$
(8)

where $R_{merg}$ represent the merged precipitation data, $n$ is the number of satellite products, $Sat_i$ represent the SPPs.

### 2.3.5 Merging criteria

In this study, daily precipitation data from four SPPs were employed to merge with RGs and explanatory variables during the summer monsoon from 2001 to 2017 over the UTAB. We developed ten models to train the DVQR, MLQR, and BMAQ

models based on nine stations (90%) and predict merged precipitation data at the target station (10%), and switch the target station with one of the training stations (Mohammadi and Aghashariatmadari, 2020).

The DVQR, MLQR, and QBMA models were optimized during the training phase to produce good modeling performance while avoiding overfitting. Hyperparameter optimization searches the optimal parameters of applied models that govern their performance (Abdalla et al., 2021). The best quantile level was selected to predict accurate and realistic merged precipitation

data by testing five quantile levels $(0.05, 0.25, 0.50, 0.75, 0.90)$ for DVQR and MLQR models, while the BMAQ model was tested using mean quantile, and $(0.1, 0.5, and\ 0.9)$.

### 2.3.6 Performance evaluation

Several descriptive and categorical statistics were employed to assess the performance and robustness of the DVQR, MLQR, and BMAQ models in terms of merging multiple SPPs over the rugged topography (UTAB). The selected descriptive

statistics include correlation coefficient (CC), Kling Gupta Efficiency (KGE), Nash-Sutcliffe efficiency (NSE), mean absolute error (MAE), and root mean square error (RMSE). CC and KEG to measure the agreement between SPPs and RGs data, the range of CC and KEG from 0 to 1, whereas 1 indicates perfect match and 0 indicates no agreement. The NSE is a technique for determining the relative magnitude of SPPs compared to RGs, and it is widely used to assess the accuracy of a hydrological simulation (Nash and Sutcliffe, 1970). However, NSE was recently used to evaluate the precipitation data (Lu





et al., 2019); it ranges from $-\infty$ to 1, whereas 1 indicates high credibility and good quality of SPPs, and a value less than 0 indicates the SPPs not credible and has low quality. The MAE and RMSE measure the mean error of SPPs while the perfect values close to 0.

$$CC = \frac{\sum_{i=1}^{n}(S_i - \bar{S})(O_i - \bar{O})}{\sqrt{\sum_{i=1}^{n}(S_i - \bar{S})^2 \sum_{i=1}^{n}(O_i - \bar{O})^2}}, \tag{9}$$

$$KGE = 1 - \sqrt{(CC - 1)^2 + (\frac{cd}{rd} - 1)^2 + (\frac{cm}{rm} - 1)^2}, \tag{10}$$

$$NSE = 1 - \frac{\sum_{i=1}^{n}(O_i - S_i)^2}{\sum_{i=1}^{n}(O_i - \bar{O})^2}, \tag{11}$$

$$PBIAS = \frac{\sum_{i=1}^{n}(S_i - O_i)}{\sum_{i=1}^{n}(O_i)}, \tag{12}$$

$$MAE = \frac{1}{n}\sum_{i=1}^{n}|S_i - O_i|, \tag{13}$$

$$RMSE = \sqrt{\frac{1}{n}\sum_{i=1}^{n}(S_i - O_i)^2}, \tag{14}$$

where $S$ is satellite estimates precipitation and $O$ represent the precipitation at RGs at day, station $i$; the $\bar{S}$ and $\bar{O}$ represent the
average precipitation from SPPs and RGs, respectively; $n$ the whole day of the study period. The $cd$ and $rd$ represent the average precipitation data, while $cm$ and $rm$ are the standard deviation of SPPs and RGs, respectively.

Additionally, we employed different categorical statistics to assess the capability of original and merged SPPs in capturing varied precipitation events, including the probability of detection (POD), false alarm ratio (FAR), frequency bias index (FBI), and critical success index (CSI). Successful detection of precipitation events ought to have POD, CSI, and FBI values
of 1 and a FAR value of 0, which have the definitions found in Eqs. (15), (16), (17), and (18), respectively.

$$POD = \frac{H}{H + M}, \tag{15}$$

$$FAR = \frac{F}{F + H}, \tag{16}$$

$$FBI = \frac{H + F}{H + M}, \tag{17}$$

$$CSI = \frac{H}{H + M + F}, \tag{18}$$

where $H$ represents the precipitation events captured by the RGs and the original and merged SPPs at the same time, $M$ represents the precipitation events captured by the RGs but not by the original and merged SPPs, and $F$ means the precipitation events captured by the original and merged SPPs but not by the RGs.



The categorical skill statistics were employed for five classes of precipitation intensity, including no precipitation ([0, 1) mm/day), light precipitation ([1, 5) mm/day), moderate precipitation ([5, 10] mm/day), heavy precipitation ([10, 25] mm/day), and extreme precipitation (25 mm/day) as shown in Table 2 (Amjad et al., 2020).

< Table 3, inserted here, please>

## 3 Results and discussion

### 3.1 Downscaling evaluation

This study applied the Bilinear (BIL) interpolation technique to downscale SPPs and explanatory variables from coarse spatial resolution (0.1º and 0.25º) to fine resolution (0.01º) to reduce the unbalance scale between pixel and rain gauge point. We evaluated the performance of SPPs before and after the interpolation step against the RGs data to check if the interpolation techniques improved the quality of the original SPPs data. The results presented in Table 3 show that the CC and PBIAS of downscaled SPPs range from 0.36 to 0.44 and from -8.1 to 13.3, respectively. In contrast, the CC and PBIAS value of original SPPs range from 0.34 to 0.43 and from -8.4 to 16.8, respectively. These results indicate that the BIL interpolation technique has little influence on improving the original SPPs data. The spatial pattern distribution of mean annual precipitation data of original and downscaled SPPs is shown in Fig. 5. Hence, the downscaling step offers a solid data foundation for training and testing for a later stage of the merging approach (Chen et al., 2018). Similar studies reported that the RGs correlated better with downscaled SPPs with BIL interpolation techniques than the original SPPs (Ulloa et al., 2017; Gebremedhin et al., 2021; Al-Dousari et al., 2008).

< Table 3, inserted here, please>

<Figure 5., inserted here, please>

### 3.2 Spatial distribution of monsoon precipitation

A critical factor in evaluating the abilities of satellite based-precipitation products (SPPs) is the characterization of spatial heterogeneity of precipitation data (Haile et al., 2009). Precipitation intensity and pattern distribution are influenced by the terrain's elevation and other interrelated elements like slope and aspect (Haile et al., 2009). The minimum monsoon precipitation is 256, 290, 300, and 324 mm, while the maximum monsoon precipitation data is 817, 1014, 1250, and 1384 mm for PERSIANN, TAMSAT, IMERG, and CMORPH, respectively, as shown in Fig. 5e-h. The mean monsoon precipitation in UTAB ranges from 337 to 928 mm from 2001 to 2017, with a decreasing pattern distribution from southwest to northeast, depending on the 10 RGs used in this study. This pattern was consistent with the results reported by (Abebe et al., 2020; Gebremicael et al., 2019). The spatial pattern distribution maps of mean monsoon precipitation of downscaled SPPs (Fig. 5e-h) indicate over/underestimation of precipitation data as compared to RGs in Fig. 6a. In particular, TAMSAT, IMERG, and CMORPH products are remarkable with overestimation precipitation data; while PERSIANN product characterized with underestimation of precipitation data during the summer monsoon. The spatial pattern distribution





produced by SPPs varies significantly from that of RGs. As a result, the SPPs are incapable of capturing the large spatial
scale attributes of the seasonal mean precipitation pattern distribution. Similar findings were reported by (Gebremicael et al.,
2019; Fenta et al., 2018) in the Nile River Basin.

Additionally, we compared the spatial distribution produced by various merging approaches present in Fig. 6b-f. The
minimum mean monsoon precipitation is 443, 479, 480, 250, and 243 (mm), while the maximum mean monsoon
precipitation is 851, 804, 780, 953, and 1013 (mm) for DVQR, BMAQ, MLQR, SMA, and OORA models, respectively. We
observed that both traditional merging methods (SMA and OORA) produced the amount of precipitation with
underestimation and overestimation across the northeastern and southwestern parts of the region, respectively. Yumnam et
al. (2022) reported that the SMA and OORA methods produced an overestimation of precipitation during the summer
monsoon. Therefore, adopting another emerging approach is necessary because traditional methods failed to produce
satisfactory precipitation during summer monsoons over the UTAB. Compared to RGs, the results obtained by the DVQR
model performed better than BMAQ and MLQR models at capturing monsoon precipitation's magnitude and spatial
variability. Overall, the merged precipitation is uniform with RGs, acknowledging the efficiency and reliability of the
applied merging approaches.

<Figure 6, inserted here, please>

### 3.3 Overall performance merged precipitation data

The present study proposed to merge multiple SPPs with RGs and explanatory variables during the summer monsoon from
2001 to 2017 over the UTAB, as described in the methods section. Fig. **7** shows the boxplot of descriptive statistics
distributions of original SPPs and merged precipitation data with traditional merging methods and quantile regression
models against the RGs. The main criteria of the boxplot divide the dataset into four items based on maximum, minimum,
median, and two quartiles, whereas the median, which divides the statistical data into two equal portions, is indicated by the
middle horizontal line. The mean CC values of downscaled SPPs characterize by 0.44, 0.44, 0.43, and 0.36 for IMERG,
CMORPH, TAMSAT, and PERSIANN, respectively. The results indicate that the daily precipitation data of downscaled
SPPs have poor performance (CC < 0.5) against RGs. The findings are also in line with Gebremicael et al. (2019), who
stated poor performance of these SPPs at the daily temporal scale in the UTAB. Moreover, the SMA and OORA methods
performed better than downscaled SPPs, whereas the CC is 0.49 and 0.47, respectively. Also, Shen et al. (Shen et al., 2014)
observed the same result during the summer season. They stated that the SMA and OOR methods are more accurate than
individual SPPs over the Tibetan Plateau. The CC of the three quantile regression models is 0.49, 0.50, and 0.50 for DVQR,
BMAQ, and MLQR models, respectively. Overall, quantile regression models' CC of merged precipitation data is higher
than traditional merging methods (SMA and OOR) and individual downscaled SPPs.

The merged precipitation data by quantile regression models recorded the lowest MAE and RMSE than downscaled SPPS,
as shown in Fig. 7b and c. The DVQR model indicates lower MAE, whereas the BMAQ model indicates lower RMSE. In
addition, based on the distribution of PBIAS in Fig. 7d, the PERSIANN is characterized by underestimation, while IMERG,





CMORPH, and TAMSAT are characterized by overestimating precipitation data. Some studies reported that the IMERG, CMORPH, PERSIANN, and TAMSAT products have significant errors over the Nile River Basin (Abebe et al., 2020; Belete et al., 2020). Among the merging approach, the DVQR model shows the lowest PBIAS flowed by SMA, BMAQ,

MLQR, and OORA models. The merged precipitation data by the DVQR model generally showed better performance than BMAQ and MLQR models. The results showed the benefits of coupling the meteorological and land surface variables with downscaled SPPs to reduce relative errors (Chao et al., 2018; Kumar et al., 2019; Nguyen et al., 2021a).

<Figure 7, inserted here, please>

Fig. 8 shows the scatter plots of downscaled SPPs and merged precipitation data based on traditional merging methods and

quantile regression models against the RGs at a daily temporal scale during the summer monsoon for the whole period. The NSE value of downscaled SPPs is less than 0.3, which is regarded as unsatisfactory (Sen Gupta and Tarboton, 2016), whereas the KGE is less than 0.6 for all downscaled SPPs. The merged precipitation data in Fig. 8e-i is relatively close to the 1:1 line, whereas the downscaled SPPs, as in Fig. 8a-d, exhibit the most scattered distribution of precipitation data, indicating that quality after merging is changed for the better. Furthermore, when it comes to merging approaches, the

quantile regression models fit better than traditional merging methods and downscaled SPPs. On the other hand, the KGE values are 0.744, 0.749, 0.771, 0.657, and 0.785, while the NSE values are 0.501, 0.484, 0.543, 0.617, and 0.615 for the OORA, SMA, MLQR, BMAQ, and DVQR models, respectively. The NSE value greater than 0.5 is considered satisfactory (Sen Gupta and Tarboton, 2016); results suggest that the quantile regression models have significantly improved the accuracy of downscaled SPPs. This result is in agreement with previous findings, such as Ma et al. (2018) and Rahman et al.

(2020a), which indicated that the BMA approach outperformed traditional merging methods. Likewise, the merged precipitation data by the DVQR model was strongly correlated with the RGs over the UTAB, which was more abundantly clear than in MLQR and BMAQ models, as seen in Fig. 8i.

<Figure 8, inserted here, please>

Table 4 shows the overall performance of downscaled SPPs, traditional methods, and quantile regression models against

RGs during the summer monsoon from 2001 to 2017 over the UTAB. The CC of downscaled SPPs is 0.66, 0.71, 0.72, and 0.75 for PERSIANN, TAMSAT, CMORPH, and IMERG, respectively, while for traditional merging methods are 0.76 and 0.77 for OOAR and SMA methods, respectively. However, the quantile regression models outperformed traditional merging methods and downscaled SPPs, whereas CC values are 0.79, 0.79, and 0.80 for MLQR, BMAQ, and DVQR models, respectively. The results indicate that all merged precipitation data have remarkable linear correlations with the RGs than the

downscaled SPPs. In addition, the quantile regression models notably with high accuracy than merged precipitation data by traditional methods and downscaled SPPs. The MAE and RMSE of original SPPs decrease by traditional merging methods (OOAR and SMA) and quantile regression models (MLQR, BMAQ, and DVQR), while the DVQR model observed with lower MAE and RMSE than other merging models. Overall, the PBIAS of downscaled SPPs indicates 15.81%, -4.74%, -8.43%, and -6.61%; for PERSIANN, TAMSAT, CMORPH, and IMERG, respectively. The significant negative and positive

PBIAS of precipitation products inputs to hydrological models increase streamflow uncertainty and fail to capture peak flow





(Reda et al., 2022; Gebremicael et al., 2022). The PBIAS of SMA and OORA methods, characterized by -0.99% and 9.24%, respectively, indicate that the SMA method improved the estimation of precipitation data. Generally, the SMA method showed the smallest PBIAS than the OORA method and downscaled SPPs (Shen et al., 2014; Yumnam et al., 2022). Regarding quantile regression models, the PBIAS of DVQR, MLQR, and BMAQ models are 0.96%, -2.94%, and -6.61%,

respectively. The DVQR model generally reduces the large error of downscaled SPPs and notably performs better than MLAR and BMAQ models.

< Table 5, inserted here, please>

Generally, the DVQR model shows better performance among the merging approaches with significant improvements in all metrics. The Taylor diagram was used to evaluate the performance of merging approaches to further debits of a

comprehensive evaluation of accuracy. Based on the CC, centered RMSE, and standard deviation (SD) statistical metrics, the Taylor diagram quantified the degree of correspondence between RGs and estimated precipitation (Wang et al., 2021). The closest points of estimated precipitation to the point of RGs represent the best accuracy. In addition, the Taylor diagram is a highly useful tool for analyzing the meteorology dataset for comparing the performance between different datasets (Chao et al., 2021). Fig. 9 shows the Taylor diagram of various precipitation sources, including downscaled SPPs, traditional merging

methods, and quantile regression models during the summer monsoon over the UTAB. The IMERG product exhibit the highest performance than other downscaled SPPs during summer monsoon over the UTAB. The merged precipitation data outperformed the downscaled SPPs over the UTAB.

Moreover, the BMAQ and DVQR models show higher performance than others in terms of CC and RMSE; however, the BMAQ model indicates a lower SD than other models. The BMAQ and DVQR models outperformed all downscaled SPPs

(IMERG, CMORPH, TAMSAT, and PERSIANN) according to CC, RMSE, and SD across the UTAB. In line with our recent findings, Ma et al. (2018) demonstrated that the merged precipitation data-based BMA model performed better than the IMERG product across the Tibetan Plateau. Also, Yumnam et al. (2022) reported that the bias-corrected-based BMAQ model outperformed IMERG product-based RMSE and SD over the complex topography.

<Figure 9, inserted here, please>

Nevertheless, the nonlinear DVQR model shows high potential capability in merging SPPs than the linear quantile regression (MLQR model) over the UTAB. In general, our findings are consistent with previous research (Abdallah et al., 2022; Niemierko et al., 2019; Nguyen et al., 2021b; Wu et al., 2022); they reported which conditioned factor is located in the middle of the distribution, where the linear quantile regression is suitable. Nevertheless, it is inappropriate to figure out conditional quantiles scattered beyond the center range. However, the D-vine copula approaches provide a way to forecast

highly nonlinear conditionally of the quantiles at the tails.

### 3.4 Monthly scale assessment

To show how the DVQR model improves the quality of merged precipitation data, we also evaluate the degree-of-fit of the daily precipitation data for each month during the summer monsoon over the UTAB. Fig. 10 shows the group of Taylor





diagram to compare the quality of different precipitation sources according to CC, RMSE, and SD during June, July, August,
and September.  Among the downscaled SPPs, the IMERG product showed the highest performance than other products
during June, July, and August, while the CMORPH product outperformed during September (Gebremicael et al., 2019). In
general, the merged precipitation data by quantile regression models and traditional merging methods outperforms the
downscaled SPPs across the UTAB. Interestingly, the merged precipitation data by traditional merging methods, including
SMA and OORA, indicates better performance next to quantile regression models during all summer monsoon months, as
seen in Fig. 10.

In addition, the results in Fig. 10 confirmed that the quantile regression models have high potential and are more capable of
merging precipitation data during each month of summer monsoon across rugged topography like the UTAB. Among the
quantile regression models, the DVQR model outperforms the BMAQ and MLQR models based on the highest CC, lower
RMSE, and close SD to RGs. In particular, as we showed in Fig. 10a-d, the CC of the DVQR model is the highest,
characterized by 0.68, 0.74, 0.71, and 0.75, while the CC of the BMAQ model values is 0.67, 0.74, 0.71, and 0.74 during
June, July, August, and September, respectively. Overall, The DVQR model showed an equivalent capability and
effectiveness to the BMAQ model based on the Taylor diagram, but it was far more capable than the MLQR model and
traditional merging methods.

<Figure 10, inserted here, please>

Fig. 11 shows the cross-validation of original SPPs and merged precipitation data against the RGs for each month during the
summer monsoon from 2001 to 2017 over the UTAB. The DVQR model has the smallest PBIAS close to zero across July,
August, and September than other merging models and downscaled SPPs products. The BMAQ model shows the smallest
PBIAS during June; also, we can denote that it has the highest bias during September, increasing the overestimation of
precipitation data. In other words, traditional merging methods and quantile regression models dramatically increase the
monthly NSE of merged precipitation data compared to downscaled SPPs, as shown in Fig. 11b. The NSE value of
downscaled SPPs range from -0.30 to 0.32 is regarded as unsatisfactory. In contrast, the traditional merging methods range
from 0.27 to 0.39, which is less than 0.5 is also indicated as unsatisfactory (Sen Gupta and Tarboton, 2016). Moreover, in
terms of quantile regression models, the NSE value of the MLQR model is 0.41, 0.34, 0.41, and 0.49, while for the BMAQ
model is 0.43, 0.54, 0.50, and 0.38; and for the DVQR model is 0.42, 0.50, 0.49, and 0.53, for June, July, August, and
September, respectively. The results indicate that the BMAQ and DVQR models improved precipitation data quality (NSE)
during July, August, and September by higher than 0.50, which is considered satisfactory. The results suggest that adding
additional explanatory variables, such as wind speed and surface soil moisture, to quantile regression models can
significantly reduce the uncertainty of downscaled SPPs (Kumar et al., 2019; Chao et al., 2018).

Furthermore, the KGE for traditional merging methods and quantile regression models is improved to 0.64–0.72 and 0.68–
0.75, respectively, when compared with the downscaled SPPs (approximately 0.47–0.71) as shown in Fig. 11c. Suggesting
that the two merging approaches bring the quality of downscaled SPPs to a great level. Among the quantile regression
models, the DVQR model exhibits a higher KGE value than BMAQ and MLQR models during all summer monsoon





months. Some previous research demonstrated that the BMAQ model has the highest potential to improve the estimation of downscaled SPPs than traditional margin methods (Yumnam et al., 2022; Rahman et al., 2020a; Ma et al., 2018). However,

our results indicate that the DVQR model has higher accuracy than the BMAQ model during the summer monsoon. This reflects the capability and robustness of high-dimensional (10-D) vine copula to capture nonlinear relationships among the input variables (Nguyen et al., 2021b; Sharifi et al., 2019; Wu et al., 2022). Overall, the performance of statistical metrics, including PBIAS, NSE, and KGE, during July and September was better than in June and August over the UTAB.

<Figure 11, inserted here, please>

**3.5 Precipitation detection assessment**

Fig. 12 shows the detection of precipitation amount based on the different intensities of downscaled SPPs, traditional merging methods, and quantile regression models during the summer monsoon over the UTAB. Fig. 12a shows the POD decrease by increasing the precipitation intensity for all the data. The POD denotes binary response estimations rather than continuous target estimations. The CMORPH product has the maximum POD among the original SPPs, whereas the

PERSIANN product has the lowest across all precipitation intensities. In traditional merging methods, the SMA method showed higher POD than the OORA method and downscaled SPPs across all precipitation intensities (Rahman et al., 2020a). The BMAQ model indicates the highest POD during light precipitation, the DVQR model shows the highest POD during moderate precipitation, and the MLQR model outperforms during heavy and extreme precipitation. The quantile regression models exhibit higher POD than traditional methods and original SPPs, which are in agreement with recent previous studies

(Ma et al., 2018; Rahman et al., 2020c). Which reported that the POD of traditional merging methods (SMA and OORA) and the BMA model were higher than single-downscaled SPPs

Fig. 12b shows the CSI of all precipitation intensities for downscaled SPPs, traditional merging methods, and quantile regression. Similarly, the CSI is the same as POD, which decreases by increasing the precipitation intensity for all precipitation data across the UTAB. Among the downscaled SPPs, the CMORPH exhibits the highest CSI during light

precipitation, while the IMERG product indicates the highest CSI during the other precipitation intensities. In other words, the SMA method shows higher CSI than the OORA method across all the precipitation intensities in traditional merging methods during the summer monsoon. Furthermore, among the quantile regression models, the DVQR model exhibits high CSI, while the MLQR model performed better for extreme precipitation greater than 25 mm. However, the BMAQ model has the lowest CSI across all precipitation intensities.

<Figure 12, inserted here, please>

Fig. 12c shows the FAR of all precipitation intensities for original SPPs, traditional merging methods, and quantile regression. The FAR increased by increasing the precipitation intensities. Among the downscaled SPPs, the IMERG product indicates the lowest FAR. At the same time, the CMORPH has the lowest expected precipitation of greater than 25 mm. In traditional merging methods, the OORA method outperforms the SMA method, which exhibits the lowest FAR across all the

precipitation intensities. The DVQR model is characterized by the lowest FAR for light and moderate precipitation, while





the BMAQ model shows the lowest CSI for heavy and extreme precipitation data. A similar result was reported by Yumnam et al. (2022), who observed that the BMAQ model is characterized by lower FAR due to less false of heavy and extreme precipitation.

In terms of showing the underestimation and overestimation of detecting precipitation intensity, the FBI was used, as seen in
Fig. 12d. The FBI changed from overestimation to underestimation by increasing the precipitation intensities. The IMERG product showed better FBI with very light overestimation to underestimation, followed by CMORPH and TAMSAT products among the downscaled SPPs. At the same time, PERSIANN exhibits high underestimation during heavy and extreme precipitation. The OORA method characterizes by the lowest overestimation of precipitation than the SMA method during light and moderate precipitation. In contrast, the SMA method indicates the lowest underestimation of precipitation
than the OORA method during heavy and extreme precipitation. The DVQR model is more capable of detecting light precipitation than other quantile regression models, while the MLQR model showed the lowest FBI than other models during all precipitation intensities except during light precipitation. The downscaled SPPs, including IMERG and CMORPH products, outperformed the two traditional merging methods and quantile regression models across all precipitation intensities over the UTAB. Generally, the results of POD, CSI, FAR, and FBI across different precipitation intensities
showed the benefits of merging the individual SPPs with RGs to reduce uncertainty and improve the detection of precipitation events. Also, the result reflects the advantage of using explanatory variables (DEM, ASP, SLP, HSHD, WS, and SSM) to improve the detection of precipitation events of downscaled SPPs (Ma et al., 2018; Kumar et al., 2019).

### 3.6 Sensitivity analysis of merged SPPs using the DVQR model

The sensitivity analysis was carried out in this section to examine the accuracy of merged precipitation data using the DVQR
model based on different quantile levels. Fig. 13 shows the radar plots of statistical metrics of mean merged precipitation data against the RGs during summer monsoon over the UTAB. The results reflected that the quality of merged precipitation data varies across the quantile levels. For example, the q0. 5, q0.75, and q0.9 notably have the highest CC while the q0.05 has the lowest CC, as shown in Fig.13a. The median quantile level (q0.5) is remarkable with positive NSE regarded as a satisfactory and q0.75 characterize with zero. In contrast, other quantile levels are characterized by high negative NSE as
seen in Fig. 13b; however, q0.05 and q0.90 are marked by the highest error.

In comparison, the median quantile level (q0.5) indicates the lowest error in MAE and RMSE, as shown in Fig. 13c and d. Overall, the q0.5 level is remarkable, with the highest accuracy (CC and NSE) and lowest error (MAE and RMSE) than other arbitrary quantiles. The results of estimating merged daily precipitation data across different quantile levels reflected the sensitivity of the DVQR model.

535                               <Figure 13, inserted here, please>





## 4 Conclusion

SPPs are reasonable alternatives with massive advantages over RGs. However, their applications are restricted due to insufficient quality compared to RGs at local and regional scales. The purpose of the present study was to merge multiple SPPs with RGs and coupling with explanatory variables using three quantile regression (DVQR, MLQR, and BMAQ) models and two traditional merging methods (SMA and OORA). Furthermore, the study provided insight into the capability and effectiveness of the DVQR model in merging multiple SPPs over the rugged topography basin. Below are highlighted findings from the present study:

(1) Downscaled SPPs cannot capture the large spatial scale attributes of the seasonal mean precipitation pattern distribution.

(2) The DVQR model improved the spatial pattern distribution of precipitation and indicated higher capabilities than BMAQ and MLQR models in terms of capturing the magnitude and spatial variability of monsoon precipitation over the rugged topography (UTAB).

(3) Cross-validation clearly shows that both the quantile regression models and traditional merging methods improved the estimation of daily precipitation data; however, all quantile regression models exhibit the highest accuracy.

(4) Based on a monthly analysis, the DVQR model outperformed BMAQ, MLQR, SMA, and OOAR models during June, July, August, and September. According to descriptive statistics, the performance of merged precipitation data during July and September was better than in June and August over the UTAB.

(5) Regarding POD and FAR, the DVQR merging approach does not significantly outperform the BMAQ, and MLQR approaches, but it has the best CSI and FBI across all precipitation intensities.

(6) The quantile level in the DVQR model is a sensitive parameter in predicting merged precipitation data, whereas the median quantile levels (q0.5) indicate lower uncertainty than other quantile levels.

In conclusion, this study advances our awareness of merging multiple satellite-based precipitations using different approaches with rain gauges and explanatory variables over the rugged topography. However, the approach has several limitations. First, the quality and quantity of rain gauges are poor, with many data gaps. Second, the fixed values of DEM, ASP, SLP, and HSHD may affect the data-driven vine copula technique's predicting ability. Third, the D-vine copula structure and ONC family may not wholly replicate the complex dependencies between variables in the real world.

The suggested DVQR model of merging SPPs with RGs and explanatory variables could improve the accuracy and spatial pattern distribution and reduce the uncertainty of estimated daily precipitation over the UTAB. This research is essential for enhancing precipitation estimation from multiple SPPs, especially in basins with sparse and unevenly distributed RGs and rugged topography such as the UTAB. Further research can focus on comparing the DVQR model to other precipitation merging approaches, such as machine learning techniques based on different temporal scales and climate conditions. It can also measure their capabilities in hydrological simulations and extreme events analysis.



**Data availability**

DEM was obtained from Shuttle Radar Topography Mission which can access from https://earthexplorer.usgs.gov/, wind speed (WS) was obtained from ERA5 (European Center for Medium-Range Weather Forecasts-ECMWF) can be downloaded from (https://cds.climate.copernicus.eu/cdsapp#!/home), while surface soil moisture (SSM) was obtained from Global Land Evaporation Amsterdam Model-GLEAM 3.6a is available in https://www.gleam.eu/. The data of CMORPH-CRD was accessed from https://www.ncei.noaa.gov/data/cmorph-high-resolution-global-precipitation-estimates/, IMERG

V06 was downloaded from https://disc.gsfc.nasa.gov/, PERSIANN-CDR is available in https://chrsdata.eng.uci.edu/, and TAMSAT v3.1 was downloaded from http://www.tamsat.org.uk/data. For the rain gauge data, it is not publicly available but can be obtained from the Ethiopian National Meteorological Agency website http://www.ethiomet.gov.et/. Three quantile regression models was conducted in R Programming Language using *vinereg* package (https://tnagler.github.io/vinereg/), *quantreg* package (https://cran.r-project.org/web/packages/quantreg/ ) and *BMA* package(https://cran.r-

project.org/web/packages/BMA).

**Author contribution**

**MA:** Conceptualization, Data curation, Formal analysis, Investigation, Methodology, Resources, Software, Visualization, Writing – original draft. **KZ:** Conceptualization, Investigation, Methodology, Funding acquisition, Supervision, Writing –

review & editing. **LC:** Validation, Writing – review & editing. **AO:** Validation, Writing – review & editing. **KH**: Validation, Writing – review & editing. **KWR:** Data curation, Validation. **LL:** Validation, Visualization. **TLT:** Validation, Visualization. **OMN:** Validation, Visualization.

**Competing interests**

The authors declare that they have no known competing financial interests or personal relationships that could have appeared to influence the work reported in this paper.

**Acknowledgments**

This study was supported by the National Natural Science Foundation of China (51879067, 51909061), Fundamental

Research Funds for the Central Universities of China (B200204038), National Key Research and Development Program of China (2018YFC1508101), Natural Science Foundation of Jiangsu Province (BK20180022), Hydraulic Science and Technology Plan Foundation of Shaanxi Province (2019slkj-B1), Hydraulic Science and Technology Plan Foundation of Jiangsu Province (2018055), and Six Talent Peaks Project in Jiangsu Province (NY-004). We also thank the Ethiopian National Meteorological Agency (NMA) for providing the precipitation data.




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







**Figure 1.** Map of the study area (Upper Tekeze Atbara Basin) and location of the rain gauge stations.







**Figure 2.** Spatial pattern distribution of explanatory variables include (a) elevation, (b) slope, (c) aspect, (d) hill-shade, (e) average wind speed, and (f) average surface soil moisture during summer monsoon from 2001 to 2017, over the UTAB.





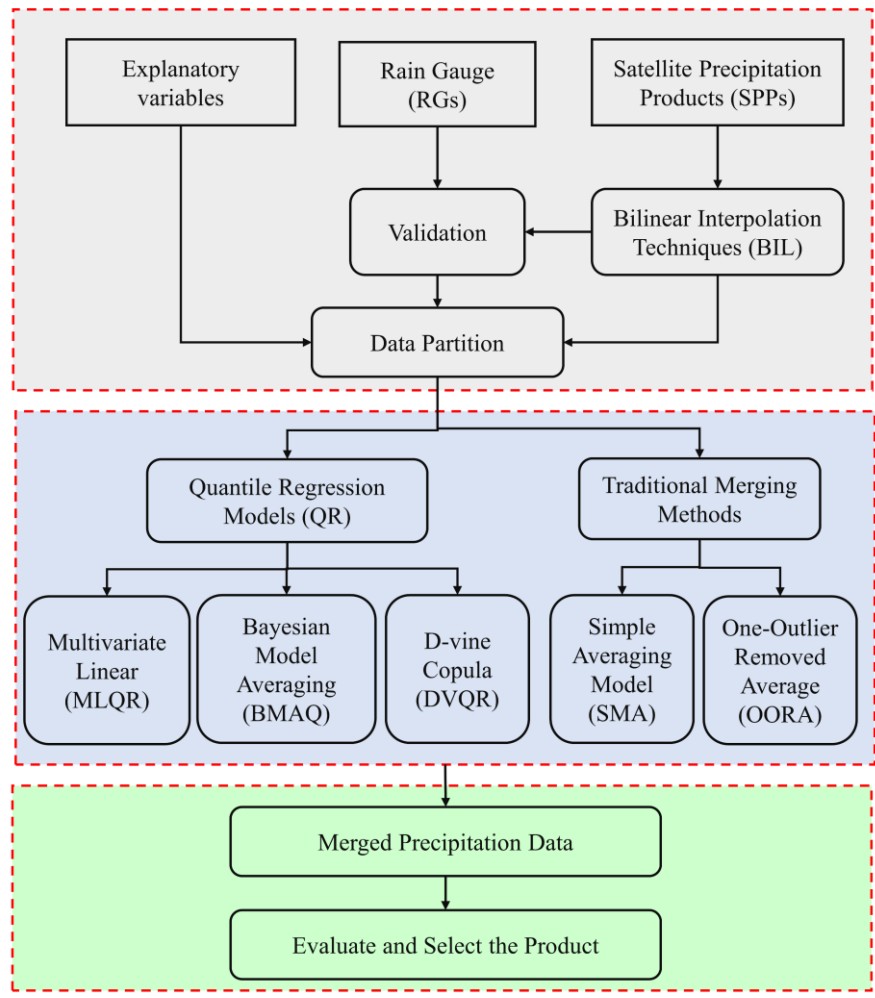

**Figure 3.** The flowchart of merging multiple SPPs with RGs coupled with explanatory variables using quantile regression models and traditional merging methods during summer monsoon over the UTAB.







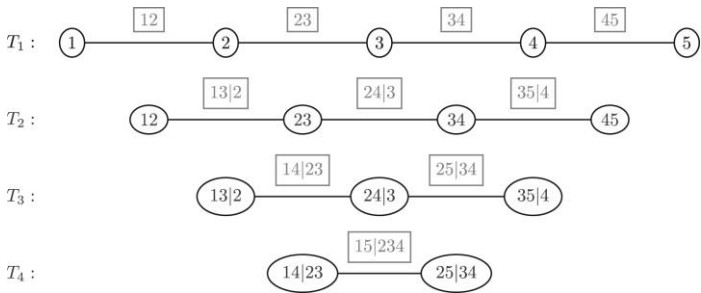

**Figure 4.** Five-dimensional D-vine copula structure based on five variables, four trees, and ten edges.


**Figure 5.** Spatial pattern distribution of mean monsoon precipitation during 2001-2017 over the UTAB for (a-d) original SPPs with coarse spatial resolution and (e-h) downscaled SPPs by BIL interpolation technique for IMERG, CMORPH, TAMSAT, and PERSIANN, respectively.








**Figure 6.** Spatial pattern distribution of mean monsoon precipitation (in mm) from 2001 to 2017, for (a) Rain Gauge, (b) DVQR, (c) BMAQ, (d) MLQR, (e) SMA, and (f) OORA over the UTAB.





**Figure 7.** Boxplot distribution of CC, MAE, RMSE, and PBIAS of individual downscaled SPPs and merged precipitation data based on traditional merging methods and quantile regression models during summer monsoon from 2001 to 2017 over the UTAB.





**Figure 8.** Comparison of average basin precipitation data of individual downscaled SPPs; (a) PERSIANN, (b) TAMSAT, (c) CMORPH, and (d) IMERG, traditional merged methods; (e) OORA and (f) SMA, and quantile regression models; (g) MLQR, (h) BMAQ, and (i) DVQR during summer monsoon from 2001 to 2017 over the UTAB.



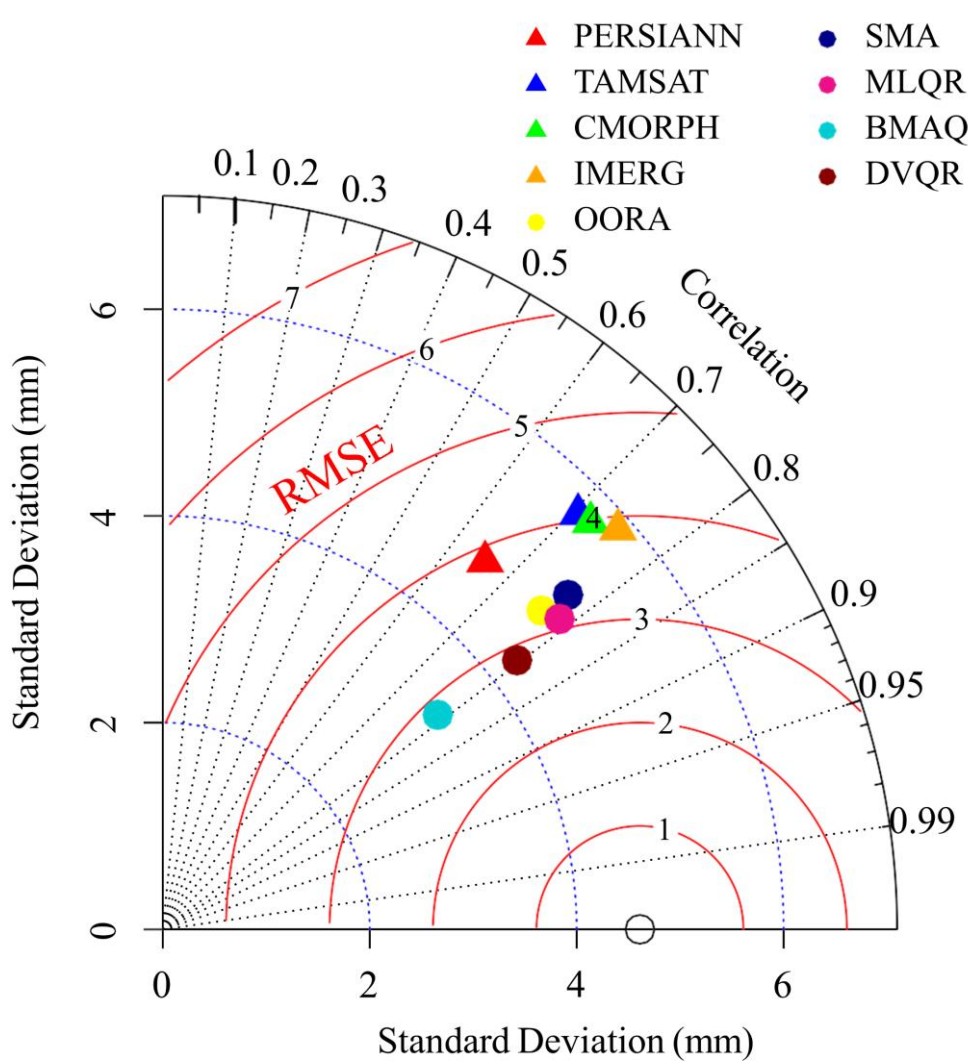

880

**Figure 9.** Taylor diagram of downscaled SPPs including PERSIANN (red triangle), TAMSAT (blue triangle), CMORPH (green triangle), and IMERG (orange triangle), traditional methods including OORA (yellow dot) and SMA (dark blue dot), and quantile regression models including MLQR (deep pink dot), BMAQ (aqua dot), and DVQR (brown dot)); for the whole period of summer monsoon from 2001 to 2017. The radial black dash line indicates the CC, the red line arc is RMSE, and 885 the radial blue dot arc is the standard deviation.





**Figure 10.** Taylor diagram of downscaled SPPs including PERSIANN (red triangle), TAMSAT (blue triangle), CMORPH (green triangle), and IMERG (orange triangle), traditional methods including OORA (yellow dot) and SMA (dark blue dot), and quantile regression models including MLQR (deep pink dot), BMAQ (aqua dot), and DVQR (brown dot); for the whole period of summer monsoon from 2001 to 2017; for (a) June, (b) July, (c) August, and (d) September.



**Figure 11.** Cross-validation statistical metrics of (a) PBIAS, (b) NSE, and (c) KGE for downscaled daily SPPs (PERSIANN, TAMSAT, CMOPRH, and IMERG) and merged precipitation data using traditional merging methods (OORA and SMA) and quantile regression models (MLQR, BMAQ, and DVQR) during June (green column), July (light green column), August (dark orange column), and September (red column) from 2001 to 2017 over the UTAB.



(a) Probability of Detection (POD)
(b) Critical Success Index (CSI)
(c) False Alarm Ratio (FAR)
(d) Frequency Bias Index (FBI)

Precipitation Amount (mm)

PERSIANN — TAMSAT — CMORPH — IMERG — OORA — SMA — MLQR — BMAQ — DVQR

**Figure 12.** Comparison of detection precipitation amount of downscaled SPPs including PERSIANN (deep pink line), TAMSAT (magenta line), CMORPH (dark blue line), and IMERG (light blue line), traditional methods including OORA (aquamarine line) and SMA (light green line), and quantile regression models including MLQR (yellow line), BMAQ (orange line), and DVQR (red line); for (a) POD, (b) CSI, (c) FAR, and (d) FBI during summer monsoon from 2001 to 2017 over the UTAB.





**Figure 13.** Sensitivity analysis of DVQR model throughout CC, NSE, MAE, and RMSE for merged precipitation data
estimated at different quantile levels during summer monsoon over the UTAB.







**Table 1.** Geographical locations of rain gauge observations, elevation, and statistical characteristics based on a daily scale, including min, max, mean, and standard deviation (SD) from January 2001 to December 2017 over the UTAB.

| Station Name | Latitude | Longitude | Elevation(m) | Min(mm) | Max (mm) | Mean(mm) | SD (mm) |
|---|---|---|---|---|---|---|---|
| Adigrat | 14.278 | 39.447 | 2509 | 0.0 | 86.0 | 2.8 | 6.9 |
| Adigudem | 13.16 | 39.13 | 1703 | 0.0 | 52.6 | 3.3 | 7.0 |
| Adwa | 14.181 | 38.878 | 1919 | 0.0 | 84.9 | 5.4 | 9.0 |
| Akxum | 14.134 | 38.747 | 2171 | 0.0 | 76.7 | 4.9 | 9.5 |
| Gonder | 12.521 | 37.432 | 1986 | 0.0 | 70.5 | 7.6 | 10.0 |
| Hselam | 13.35 | 39.27 | 2241 | 0.0 | 78.0 | 4.3 | 7.8 |
| Lalibela | 12.039 | 39.04 | 2419 | 0.0 | 74.0 | 5.3 | 8.3 |
| Maichew | 12.784 | 39.534 | 2433 | 0.0 | 61.0 | 3.8 | 7.7 |
| Mekele | 13.471 | 39.531 | 2249 | 0.0 | 60.2 | 3.7 | 7.3 |
| Shire | 14.102 | 38.295 | 1902 | 0.0 | 101.8 | 7.2 | 10.7 |










**Table 2.** Classification of precipitation intensity

| Name | Precipitation intensity (mm/day) |
|---|---|
| No precipitation | < 1 |
| Light precipitation | 1 - 5 |
| Moderate precipitation | 5 - 10 |
| Heavy precipitation | 10 - 25 |
| Extreme precipitation | > 25 |










**Table 3.** Mean values of CC, MAE, RMSE, and PBIAS for original (ORI) and downscaled (BIL) SPPs by BIL interpolation techniques at the daily scale during the summer monsoon from 2001 to 2017 over the UTAB.

| Products | Method | CC | MAE | RMSE | PBIAS % |
|---|---|---|---|---|---|
| IMERG | ORI | 0.43 | 5.0 | 9.4 | -9.9 |
| | BIL | 0.44 | 4.9 | 9.0 | -9.6 |
| CMORPH | ORI | 0.43 | 5.0 | 8.9 | -8.3 |
| | BIL | 0.44 | 4.9 | 8.6 | -7.9 |
| TAMSAT | ORI | 0.41 | 5.0 | 8.6 | -8.4 |
| | BIL | 0.42 | 4.9 | 8.3 | -8.1 |
| PERSIANN | ORI | 0.34 | 4.9 | 8.5 | 16.8 |
| | BIL | 0.36 | 4.8 | 8.4 | 13.3 |









**Table 4.** Descriptive statistics (CC, MAE, RMSE, and PBIAS) of original daily SPPs (PERSIANN, TAMSAT, CMOPRH, and IMERG) and merged precipitation data using traditional merging methods (OORA and SMA) and quantile regression models (MLQR, BMAQ, and DVQR) during summer monsoon from 2001 to 2017 over the UTAB.

| Datasets | CC | MAE (mm/day) | RMSE (mm/day) | PBIAS (%) |
|----------|----|--------------|---------------|-----------|
| PERSIANN | 0.66 | 2.66 | 3.94 | 15.81 |
| TAMSAT | 0.71 | 2.74 | 4.07 | -4.74 |
| CMORPH | 0.72 | 2.58 | 3.99 | -8.43 |
| IMERG | 0.75 | 2.46 | 3.89 | -6.61 |
| OORA | 0.76 | 2.19 | 3.26 | 9.24 |
| SMA | 0.77 | 2.21 | 3.31 | -0.99 |
| MLQR | 0.79 | 2.14 | 3.12 | -6.61 |
| BMAQ | 0.79 | 2.10 | 2.85 | -2.94 |
| DVQR | **0.80** | **1.97** | **2.86** | **0.96** |