# Peer review of "A D-vine copula-based quantile regression towards merging satellite precipitation products over a rugged topography: A case study at the upper Tekeze Atbara Basin of the Nile Basin"

_Hydrology and Earth System Sciences, 2023_

## Author Response (AR1)

**Revision Notes for Manuscript Hess-2023-179**

We highly appreciated the comments from the referee and thank the editor for managing this manuscript. All the comments were addressed and answered separately; you may find our answers and the related changes about them on the following pages (blue color) and changes in the main text in red color (track changed). We have carefully considered the suggestions of the reviewers and provided a point-by-point response to the given comments.

**Response to referee comment: Anonymous Referee #1**

We sincerely thank Anonymous Referee #1 for dedicating valuable time and effort to review our manuscript and provide insightful comments. We are confident that the constructive feedback will significantly enhance our work's clarity and quality. Please find below our detailed responses to each of their comments.

**Comment:** This study presents a new study on how to apply a D-vine copulas-based quantile regression method to merge different satellite precipitation products. A comparison of this proposed new method with other existing methods was then conducted. Overall, this new method apparently shows a better performance than the other methods. To further improve the quality of this manuscript, I think that there are several issues that the authors should consider to address.

**Response:** Thank you for your positive feedback on our manuscript. Based on your comments, we have carefully analyzed the manuscript and identified the areas that need improvement. In addition, we have considered clarity, abstract, objectives, methodology, figures, and discussion to address the mentioned issues:

**Comment:** In the abstract, more quantitative results are needed.

**Response:** Thank you for your constructive comment. Some quantitative results were added to the abstract. (P2 L41-42).

**Comment:** Lines 107-115: This paragraph is important to summarize why you conduct this study and what objectives your study have. This paragraph need be rephrased. The data input information need be moved to the Methodology section.

**Response:** Thank you for your comment. We agree with you, and the mentioned paragraph has been rephrased accordingly. Also, the section on data input information has been moved to the methodology section, as the reviewer suggested.

**Comment:** Per the requirement of HESS, all figures need be inserted into the text. I suggest reorganizing this manuscript by making this methodology as a generic method. For example, a better tile should be "A D-vine copula-based quantile regression for merging satellite precipitation products over rugged topography: A case study at the upper Tekeze Atbara Basin of the Nile Basin". In addition, you need describe the methodology as a generic method first and then describe the case study area.

**Response:** Thank you for your suggestion. We inserted all the figures and tables into the text through the manuscript. Also, we agree with your suggestion for making the methodology a generic method, and the manuscript title was modified accordingly.

**Comment:** Discussion is kind of missing. The authors need provide some descriptions on the advantages, potential limitations, further studies on this topic in this discussion.

**Response:** Thank you for your suggestions. We added a discussion section including previous studies' findings and our current study's advantages. Also, we highlight the limitations of the present study and the future direction of research. (P29-31 L735-809).

**Comment:** Discussion can be combined with the conclusion section. The conclusions need be more concised and just describe the major findings.

**Response:** Thank you for your suggestions. We improved the conclusion section, and the limitation and future direction moved from the conclusion to the discussion section.

**Response to referee comment: Anonymous Referee #2**

We greatly appreciate Anonymous Referee #2 for providing valuable and constructive comments that greatly help us improve the quality of the manuscript. We have fully considered the comments and will revise the manuscript accordingly. Please find below our detailed responses to each of their comments.

**Comment:** The authors presented a data merging method to combine multiple remote-sensing rainfall products and rain gauge observation. The method was applied in a basin located in Ethiopia and its performance was compared against other data merging methods. This manuscript is generally well-designed with clear results. However, there are several major concerns that should be addressed.

**Response:** Thank you for the positive evaluation and encouraging comments on our manuscript. Based on your comments, we have carefully analyzed the manuscript and identified the areas that need improvement.

1. the impacts of different scales. The rain gauge observes rainfall at a local or point scale, while the satellite products are in km or tens of km scale. When the authors try to compare the merged products against rain gauge observation, the mismatch of scale should be discussed in more detail and carefully.

**Response:** Thank you for your comment. The mismatching between rain gauges (points) and satellite-based precipitation products (pixels) has been discussed in the first part of the discussion section. (P29 L733-742).

2. The validation. The authors take a ten-fold way to train and validate their results. However, only ten stations are available. it is suggested the authors use the other 65 stations for validation.

**Response:** Thank you for your feedback and suggestion to use additional stations for validation. Unfortunately, the other 65 stations are unavailable for validation in our study. Due to data limitations, we were constrained to use the available ten stations for training and validation. 3. The reliability of downscaled soil moisture data. the authors downscale soil moisture to 0.01 degree, but did not show the validation results. Soil moisture is a highly spatial heterogeneous variable. The authors should check and validate the downscaled soil moisture data.

**Response:** Thank you for your comment regarding validating our soil moisture data, both the original and downscaled satellite data. Unfortunately, we do not have access to observed soil moisture data for validation purposes, which is a limitation of our study. The soil moisture data was used as an explanatory variable to improve the estimation of precipitation based on the suggestion of an investigation by Kumar et al. (2019).

Kumar, A., Ramsankaran, R., Brocca, L., and Munoz-Arriola, F.: A Machine Learning Approach for Improving Near-Real-Time Satellite-Based Rainfall Estimates by Integrating Soil Moisture, Remote Sens., 11, 20, https://doi.org/10.3390/rs11192221, 2019.

**Some minor issues, like**

4. L34, what are the downscaled SPPs, from where? The abstract should be self-explained.

**Response:** Thank you for your comment. "SPPs" refers to Satellite-based Precipitation Products mentioned in L26. These products are obtained from Tropical Applications of Meteorology using SATellite (TAMSAT v3.1), the Climate Prediction Center MORPHing Product Climate Data Record (CMORPH-CDR), Global Precipitation Measurement (GPM) Integrated Multi-satellite Retrievals for GPM (IMERG v06) and Precipitation Estimation from Remotely Sensed Information using Artificial Neural Network (PERSIANN-CDR). We added the mentioned above products to the abstract. (P1 L32-35).

**5. L36, what is UTAB?**

**Response:** Thank you for your comment. We apologize for the omission of an explanation for "UTAB" at L36. UTAB stands for "Upper Tekeze-Atbara Basin," which is a study area we selected in this study. (P2 L37).

6. In the abstract part, it is better to show the results in a more quantitative way, like adding some values of the metrics used in this study.

**Response:** Thank you for your suggestion. We agree with your recommendation to enhance the abstract's clarity and informativeness by incorporating quantitative data. Some quantitative results were added to the abstract. (P2 L41-42).

7. L109, "Here in this present study", please be concise.

**Response:** Thank you for your comment. We appreciate your suggestion for conciseness in our manuscript. The above sentence has been modified to be more concise. (P4 L118).

8. L110, SPP has been defined before, do not need to do it again.

**Response:** Thank you for your comment. We removed any other repeated definitions of SPPs accordingly.

9. L148, what is the meaning of "another NOAA-CPC"?

**Response:** Thank you for your comment. "Another NOAA-CPC" refers to a category of products provided by the National Oceanic and Atmospheric Administration Climate Prediction Center (NOAA-CPC). NOAA-CPC offers various types of satellite-based precipitation products, including but not limited to the African Rainfall Climatology (ARC2) and the African Rainfall Estimation Algorithm (RFE). (P6 L204)

10. L151, what is the "PM"?

**Response:** Thank you for your comment. We apologize for the oversight in our manuscript. In L151, "PM" was written mistakenly instead of "passive microwave (PMW)." We appreciate your keen eye in identifying this error, and the corrections have been made in the manuscript's revised version. (P6 L207)